# POINTING OUT SQL QUERIES FROM TEXT

## ABSTRACT

The digitization of data has resulted in making datasets available to millions of users in the form of relational databases and spreadsheet tables. However, a majority of these users come from diverse backgrounds and lack the programming expertise to query and analyze such tables. We present a system that allows for querying data tables using natural language questions, where the system translates the question into an executable SQL query. We use a deep sequence to sequence model in wich the decoder uses a simple type system of SQL expressions to structure the output prediction. Based on the type, the decoder either copies an output token from the input question using an attention-based copying mechanism or generates it from a fixed vocabulary. We also introduce a value-based loss function that transforms a distribution over locations to copy from into a distribution over the set of input tokens to improve training of our model. We evaluate our model on the recently released WikiSQL dataset and show that our model trained using only supervised learning significantly outperforms the current state-of-the-art Seq2SQL model that uses reinforcement learning.

## 1 INTRODUCTION

The IT revolution of the past few decades has resulted in a large-scale digitization of data, making it accessible to millions of users in the form of databases and spreadsheet tables. Despite advances in designing new high-level programming languages and user interfaces, querying and analyzing such tables usually still requires users to write small programs in languages such as SQL or Excel, which is unfortunately beyond the programming expertise of a majority of end-users (Gualtieri, 2009). Thus, building effective semantic parsers that can translate natural language questions into executable programs has been a long-standing goal to improve end-user data accessibility (Poon, 2013; Zettlemoyer & Collins, 2005; Pasupat & Liang, 2015; Li et al., 2005; Gulwani & Marron, 2014).

Recent work has shown that recurrent neural networks with attention and copying mechanisms (Dong & Lapata, 2016; Neelakantan et al., 2016; Jia & Liang, 2016) can be used effectively to build successful semantic parsers. Notably, Zhong et al. (2017) recently introduced the state-of-the-art Seq2SQL model for question to SQL translation in the supervised setting, where programs are explicitly provided with their corresponding questions. The Seq2SQL model shows that using separate decoders for different parts of a query (i.e., aggregation operation, target column, and where predicates) increases prediction accuracy, and reinforcement learning further improves the model by allowing it to learn semantically equivalent queries beyond supervision.

In this paper, we present a new encoder-decoder model as an extension of the attentional seq2seq model for natural language to SQL program translation and a training approach that is capable of learning the model in an effective and stable manner. Figure 1 shows an example table-question pair and how our system generates the answer by executing the synthesized SQL program.

First, we present a simple *type system* to control the decoding mode at each decoding step (cf. Sect. 2). Based on the SQL grammar, a decoder cell is specialized to either select a token from the SQL built-in vocabulary, generate a pointer over the table header and the input question to copy a table column, or generate a pointer to copy a constant from the user's question. The type system allows us to have a fine-grain control over the decoding process while retaining the simplicity of the sequence structure, as opposed to designing multiple decoders for different language components or adding extra controllers for expansion of production rules (Krishnamurthy et al., 2017).

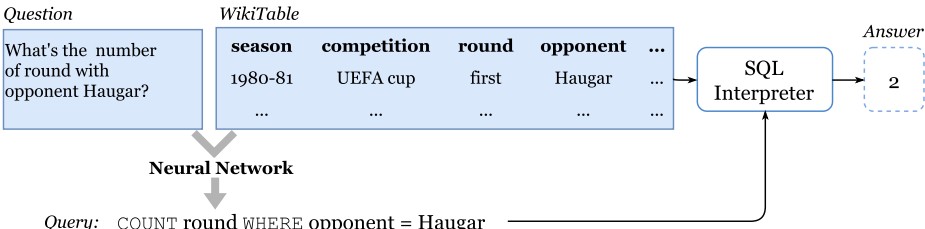

Figure 1: Answering a table question by synthesizing a query and executing it on the provided table.

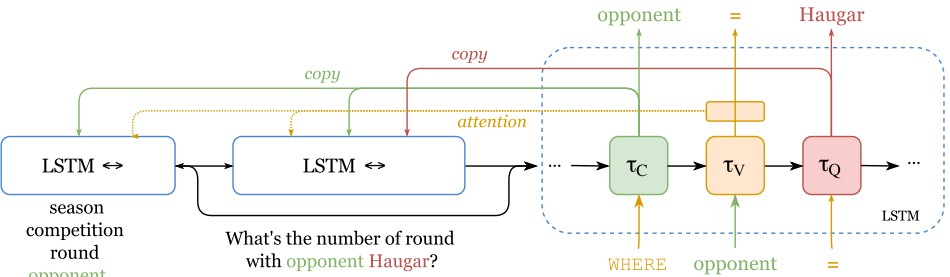

Figure 2: Model overview for the example in Figure 1. The model encodes table columns as well as the user question with a bidirectional LSTM and then decodes the hidden state with a typed LSTM, where the decoding action for each cell is statically determined.

Second, we constructed an objective function that allows us to effectively train our model to copy correct values (cf. Sect. 3). Training copying decoders can be challenging when the value to be copied appears in multiple places in the input (i.e. both in the question and the table headers). Our solution to the problem is to use a new value-based loss function that transfers the distribution over the pointer locations in the input into a distribution over the set of tokens observed in the input, by summing up the probabilities of the same vocabulary value appearing at different input indices. Our results show that our training strategy performs better than alternatives (e.g., direct supervision on pointers). Our approach is very robust and consistently converges to high-accuracy models starting from random initializations.

We have evaluated our approach on the recently released WikiSQL dataset (Zhong et al., 2017), a corpus consisting of over 80,000 natural language question and pairs. Our results in Sect. 4 show that our model can significantly outperform the current state-of-the-art Seq2SQL model (Zhong et al., 2017), without requiring a reinforcement learning refinement phase (**59.5%** vs 48.3% for exact syntactic match and **65.1%** vs 59.4% for execution accuracy). Also, with a series of ablation experiments, we analyze the influence of different components of our model on the overall results.

## 2 MODEL

We generate SQL queries from questions using an RNN-based encoder-decoder model with attention and copying mechanisms (Vinyals et al., 2015; Gu et al., 2016; Zhong et al., 2017). However, we use the known structure of SQL to *statically* determine the "type" of output of a decoding step while generating the SQL query. For example, we know from the grammar that the third token (after the aggregation function) of the query is always a column name specifying the aggregated column. Thus, when decoding, as shown in Figure 2, we statically determine the type of the token to generate based on its decoding time stamp, and then use a specialized decoder to generate the output: if we have to produce a column name or a constant, we enforce the use of a copying mechanism, otherwise we project the hidden state to a built-in vocabulary to obtain a built-in SQL operator. This means that we only need to maintain a small built-in decoder vocabulary (sized 17) for all operators.

### 2.1 ENCODER

Our encoder is a bidirectional recurrent neural network (RNN) using Long Short-Term Memory (LSTM) cells. As input tokens, we use the concatenation of the table header (i.e., the column

names) of the queried table and the user query, i.e., $X = [x_c^{(1)}, \ldots, x_c^{(C)}, x_q^{(1)}, \ldots, x_q^{(Q)}]$. This concatenation allows the model to learn how to compute a joint representation for both columns and the input query. We use $|X|$ to represent the input sequence length (equal to $C + Q$).

**Token Embedding**   To handle the large number of different tokens in the input query, we combine a pre-trained character $n$-gram embedding and a pre-trained global word embedding. For a token $x$, we compute its embedding $emb_e(x)$ as the concatenation of its word embedding and the average embeddings of all $n$-gram features contained in $x$, in the same way as Zhong et al. (2017). Formally, if $W_{\text{word}}$ is a pre-trained word model, $x[i, j]$ is the character sequence from $i$ to $j$ in $x$, $W_{n\text{-gram}}$ is a pre-trained $n$-gram for the $n$-gram feature set $V$, and $N_x$ is the number of $n$-gram features contained in the word, then

$$emb_e(x) = \left( W_{\text{word}}(x), \; \frac{1}{N_x} \sum_{\substack{1 \le i < j \le |x| \\ x[i,j] \in V}} W_{n\text{-gram}}(x[i,j]) \right).$$

We use the pre-trained $n$-gram model by Hashimoto et al. (2017) and the GloVe embedding (Pennington et al., 2014) for words; both are set untrainable to avoid over-fitting.

**Bidirectional RNN**   We feed embedded tokens into a bidirectional RNN composed of LSTM cells $\mathcal{C}_{e,fw}, \mathcal{C}_{e,bw}$, computing

$$(o_{e,fw}^{(k+1)}, h_{e,fw}^{(k+1)}) = \mathcal{C}_{e,fw}(h_{e,fw}^{(k)}, emb_e(x^{(k)})) \qquad (o_{e,bw}^{(k)}, h_{e,bw}^{(k)}) = \mathcal{C}_{e,bw}(h_{e,bw}^{(k+1)}, emb_e(x^{(k)})),$$

and will use the sequence $O_e = [o_e^{(1)}, \ldots, o_e^{(|X|)}]$ for $o_e^{(k)} = (o_{e,fw}^{(k)}, o_{e,bw}^{(k)})$ as the learned representation of token $x^{(k)}$ for the attention and copying mechanisms of our decoder. We initialize the forward encoder with hidden states $h_{e,fw}^{(0)} = \mathbf{0}$ and initialize the backward encoder with $h_{e,bw}^{(|X|)}$, the last hidden state of encoder $h_{e,fw}^{(|X|)}$.

## 2.2   TYPED DECODER

**Output Grammar**   Our model uses types abstracted from the grammar of the target language to improve the decoding performance. Concretely, we know that the subset of SQL necessary to answer WikiSQL Questions can be represented using the following grammar, in which $t$ refers to the name of the table being queried, $c$ refers to a column name in the table, and $v$ refers to any open world string or number that may be used in the query:

$$
\begin{aligned}
Q &\;\rightarrow\; s\,c\, \texttt{From}\, t\, \texttt{Where}\, p \\
s &\;\rightarrow\; \texttt{Select} \,|\, \texttt{Max} \,|\, \texttt{Min} \,|\, \texttt{Count} \,|\, \texttt{Sum} \,|\, \texttt{Avg} \\
p &\;\rightarrow\; c\, op\, v \,|\, p\, \texttt{And}\, p \\
op &\;\rightarrow\; = \,|\, > \,|\, \ge \,|\, < \,|\, \le
\end{aligned}
$$

A consequence of this observation is that we can, based on the tokens generated so far, determine the "type" of the next token to generate. For example, after generating the two tokens "`Select Id`", we know that the following token must be one of the column names from the queried table. We found it sufficient to distinguish three different cases by types:

- $\tau_{\mathcal{V}}$ The output is a token from the terminals $\mathcal{V} = \{\texttt{Select}, \texttt{From}, \texttt{Where}, \texttt{Id}, \texttt{Max}, \texttt{Min}, \texttt{Count}, \texttt{Sum}, \texttt{Avg}, \texttt{And}, =, >, \ge, <, \le, \texttt{<END>}, \texttt{<GO>}\}$ of our grammar.
- $\tau_C$ The output has to be a column name, which will be copied from either the table header or the question section of $X$. Note that the column required for the correct query may not be mentioned explicitly in the question.
- $\tau_Q$ The output is a constant that would be copied from the question section of $I$.

Since the SQL grammar can be written in regular expression form as "`Select` $s\,c$ `From` $t$ `Where` $(c\,op\,v)^*$", the output types can be described as $\tau_{\mathcal{V}}\tau_{\mathcal{V}}\tau_C\tau_{\mathcal{V}}\tau_C\tau_{\mathcal{V}}(\tau_C\tau_{\mathcal{V}}\tau_Q)^*$. We can then use the type of the output token we want to generate to specialize the decoder.

**Decoder RNN**  We use a standard RNN, based on an LSTM cell with attention over $O_e$ to generate the target program $O$. Notably, we initialize the decoder from both the final hidden states $h_{e,bw}^{(0)}, h_{e,fw}^{(|X|)}$ and the hidden states $h_{e,fw}^{(C)}, h_{e,bw}^{(C)}$ generated at index $C$, the index of the end of the table header in $X$. This state forwarding strategy allows the decoder to directly access the encoding of column names to improve decoding accuracy. Using $i_d^{(k)}$, $o_d^{(k)}$ and $h_d^{(k)}$ to denote the input (resp. output, hidden state) of the LSTM cell at decoding step $k$, we define three different output layers for our three output types:

$\tau_\mathcal{V}$  We define $u^{(k,\ell)} = v^T \tanh(W_h h_d^{(k)} + W_o o_e^{(\ell)})$ using learnable parameters $W_h$, $W_o$, $b_\mathcal{V}$ and use it compute an attention mask $\alpha^{(k)} = \text{softmax}([u^{(k,1)} \ldots u^{(k,I)}])$. The chosen output token $o_d^{(k)}$ is then computed as $o_d^{(k)} = \text{argmax}(W_\mathcal{V}(O_e \alpha^{(k)}) + b_\mathcal{V})$, where $W_\mathcal{V}, b_\mathcal{V}$ are trainable variables for $\tau_\mathcal{V}$ decoding, and $O_e \alpha^{(k)}$ is the attention vector.

Then, the input to the next decoder cell is $i_d^{(k+1)} = (emb_d(o_d^{(k)}), O_e \alpha^{(k)})$, the concatenation of the token embedding and the attention vector, where the embedding function $emb_d$ is a trainable embedding for built-in SQL operators.

$\tau_C, \tau_Q$  We use the same approach to compute the attention mask $\alpha^{(k)}$. However, instead of projecting $O_e \alpha^{(k)}$ to obtain the output, the model generates $o_d^{(k)}$ by copying a token $v$ from the input sequence $X$. The index $l$ of the token to copy is calculated by $l = \text{argmax}([\alpha^{(k,1)} \ldots \alpha^{(k,|X|)}])$, the one with the highest attention value, and the decoder output $o_d^{(k)}$ is set to $x^l$. For the $\tau_Q$ decoder, only the question part of $X$ is considered.

The input $i_d^{(k+1)}$ to the next decoder cell reuses the embedding of the copied token, and is computed as the concatenation $i_d^{(k+1)} = (emb_e(o_d^{(k)}), O_e \alpha^{(k)})$ of the token embedding and the attention vector.

As all different decoder types consume and produce similar values, they could easily be exchanged or extended if more types need to be supported. The advantage of this construction is that only a very small output vocabulary of SQL operators needs to be considered, whereas all other values are obtained through copying.

## 3  TRAINING

The model is trained from question-SQL program pairs $(X, Y)$, where $Y = [y^{(1)}, \ldots, y^{(|Y|)}]$ is a sequence representing the ground truth program for question $X$. Different typed decoder cells in our model are trained with different loss functions.

$\tau_\mathcal{V}$ **loss:**  This is the standard RNN case, i.e. the loss for an output token is the cross-entropy of the one-hot encoding of the target token and the distribution over the decoder vocabulary $\mathcal{V}$:

$$loss_\mathcal{V}(k) = -\text{onehot}(y^{(k)}) \cdot \log(\text{softmax}(W_\mathcal{V}(\alpha_\mathcal{V}^{(k)} O_e) + b_\mathcal{V})).$$

$\tau_C, \tau_Q$ **loss:**  In this case, our objective is to copy a correct token from the input into the output. As the original input-output pair does not explicitly contain any pointers, we first need to find an index $\lambda_k \in [1, \ldots, |X|]$ such that $y^{(k)} = x^{(\lambda_k)}$. In practice, there are often multiple such indices, i.e., the target token appears several times in the input query (e.g., both as a column name supplied from the table information and as part of the user question). We define two loss functions for this case and evaluate both.

- *Pointer-based loss*: We pick the smallest $\lambda_k$ with $y^{(k)} = x^{(\lambda_k)}$ and compute the loss as cross entropy between this index and the chosen index, i.e.,

$$loss_C^{\text{pntr}}(k) = -\text{onehot}(\lambda_k) \cdot \log(\text{softmax}(\alpha_C^{(k)}))$$

- *Value-based loss*: While $loss_C^{\text{pntr}}$ trains the network to generate the correct output sequence, it restricts the model to only point to the first occurrence in the input sequence. In contrast, we can allow the decoder to choose any one of the input tokens with the correct value. For that, we

define a value-based loss functions that transforms the computed distribution over locations into a distribution over the set of tokens in the input. We considered to strategies for this:

– *Max Transfer*: This strategy calculates the probability of copying a token $v$ in the input as the maximum probability of pointers that point to token $v$:

$$\phi_{\max}^{(k)}(v) \quad = \max_{1 \leq l \leq |X|} \{\alpha^{(k,\ell)} \mid x^{(l)} = v\}$$

– *Sum Transfer*: This strategy calculates the probability of copying a token $v$ in the input vocabulary as the sum of probabilities of pointers that point to token $v$:

$$\phi_{\text{sum}}^{(k)}(v) \quad = \sum_{1 \leq l \leq |X|} \{\alpha^{(k,\ell)} \mid x^{(l)} = v\}$$

For both strategies, we calculate the loss function by:

$$loss_C^{\text{val}}(k) \quad = -\operatorname{onehot}(y^{(t)}) \cdot \log([\phi^{(k)}(v) \mid v \in \mathsf{Set}(X)]).$$

When training with the sum-transfer loss function, we adapt the outputs of the $\tau_Q$ and $\tau_C$ decoder cells to be the tokens with the highest transferred probabilities, computed by $\operatorname{argmax}_{v \in X}(\phi_{\text{sum}}^{(k)}(v))$, so that decoding results are consistent with the training goal.

The overall loss for a target output sequence $O$ can then be computed as the sum of the appropriate loss functions for each individual output token $o^{(k)}$.

# 4 EVALUATION

We evaluate our model on WikiSQL dataset (Zhong et al., 2017) by comparing it with prior work and our model with different sub-components to analyze their contributions.

## 4.1 EXPERIMENT SETUP

We use the sequence version of the WikiSQL dataset with the default train/dev/test split. Besides question-query pairs, we also use the tables in the dataset to preprocess the dataset.

**Preprocessing** We first preprocess the dataset by running both tables and question-query pairs through Stanford Stanza (Manning et al., 2014) using the script included with the WikiSQL dataset, which normalizes punctuation and cases of the dataset. We further normalize each question based on its corresponding table: for table entries and columns occurring in questions or queries, we normalize their format to be consistent with the table. This process aims to eliminate inconsistencies caused by different whitespace, e.g. for a column named "country (endonym)" in the table, we normalize its occurrences as "country ( endonym )" in the question to "country (endonym)" so that they are consistent with the entity in table. Note that we restrict our normalization to only whitespace, comma (','), period ('.') and word permutations to avoid over-processing. We do not edit tokens: e.g., a phrase "office depot" occurring in a question or a query will not be normalized into "the office depot" even if the latter occurs as a table entry. Similarly, "california district 10th" won't be normalized to "california 10th", and "citv" won't be normalized to "city". We also treat each occurrence of a column name or a table entry in questions as a single word for embedding and copying (instead of copying multiple times for multi-word names/constants).

**Dataset** After preprocessing, we filter the training set by removing pairs whose ground truth solution contains constants not mentioned in the question, as our model requires the constants to be copied from the question. We train and tune our model only on the filtered training and filtered dev set, but we report our evaluation on the full dev and test sets. We obtain 59,845 (originally 61,297) training pairs, 8,928 (originally 9,145) dev pairs and 17,283 test pairs (the test set is not filtered).

**Column Annotation** We annotate table entry mentions in the question with their corresponding column name iff the table entry mentioned uniquely belongs to one column of the table. The purpose of this annotation is to bridge special column entries and their column information that cannot be

learned elsewhere. For example, if an entity "rocco mediate" in the question only appears in the "player" column in the table, we annotate the question by concatenating the column name in front of the entity (resulting in "player rocco mediate"). This process resembles the entity linking technique used by Krishnamurthy et al. (2017), but in a conservative and deterministic way.

**Model Setup** We use the pre-trained $n$-gram embedding by Hashimoto et al. (2017) (100 dimensions) and the GloVe word embedding (100 dimension) by Pennington et al. (2014); each token is embedded into a 200 dimensional vector. Both the encoder and decoder are 3-layer bidirectional LSTM RNNs with hidden states sized 100. The model is trained with question-query pairs with a batch size of 200 for 100 epochs. During training, we clip gradients at 10 and add gradient noise with $\eta = 0.3, \gamma = 0.55$ to stabilize training (Neelakantan et al., 2015). The model is implemented in Tensorflow and trained using the Adagrad optimizer (Duchi et al., 2011).

## 4.2 OVERALL RESULT

Table 1 shows the results of our model with the best performance on the dev set, compared against the augmented pointer model and Seq2SQL model (with RL), both by Zhong et al. (2017). We report both the accuracy computed with exact syntax match ($Acc_{syn}$) and the accuracy based on query execution result ($Acc_{ex}$). Since syntactically different queries can be equivalent on the table (e.g., queries with different predicate orders compared to the ground truth), the execution accuracy in all cases is higher than the corresponding syntax accuracy.

Our best model achieves 61.0% on the filtered dev set, and it is trained with our value-based loss with sum-transfer strategy. Our model's syntax accuracy (Test $Acc_{syn}$) on the test set for problems whose ground truth contains [0, 1, 2, 3, 4] predicates is [54.2%, 65.0%, 50.9%, 37.6%, 23.6%], which indicates that our model retains the ability to correctly generate long queries.

| Model | Filtered Dev $Acc_{syn}$ | Dev $Acc_{syn}$ | Dev $Acc_{ex}$ | Test $Acc_{syn}$ | Test $Acc_{ex}$ |
|---|---|---|---|---|---|
| Pointer Model | - | 44.1% | 53.8% | 43.3% | 53.3% |
| Seq2SQL | - | 49.5% | 60.8% | 48.3% | 59.4% |
| Our Model | 61.0% | **59.6%** | **65.2%** | **59.5%** | **65.1%** |

Table 1: Dev and test accuracy of the model, where $Acc_{syn}$ refers to syntax accuracy and $Acc_{ex}$ refers to execution accuracy.

## 4.3 ABLATION TESTS

While the overall results show that our model significantly improves over prior work, we now analyze different sub-components of our model individually to better understand their contribution to the overall performance. We ran four sets of abalation tests on our model, running each model 5 times. All model variances are based on the model described in Sect. 4.1 with same hyper-parameters, and the model accuracy on the (filtered) development set during training is plotted in Fig. 3.

- *Type-based decoding*: We compare our model with and without type-driven specialization of the decoder cell in Fig. 3a. For the untyped model, we directly concatenate all SQL operators in the front of table header and set all decoder cells to copy mode. The result shows that while types do not significantly improve model performance (with an average improvement 1.4%), they allow the model to stabilize within fewer epochs. Additionally, we also observed that typed decoders increase the training speed per epoch by approximately $\sim$23%.

- *Loss function*: We compare the three training objectives and corresponding decoding strategies described in Sect. 3 in Fig. 3b. The results show that the sum-transfer strategy significantly improves training stability and model accuracy compared to other strategies typically used in pointer models. Notably, while the value-based loss with max-transfer strategy outperforms the pointer-based loss in its best runs (with an accuracy of 56.4%), its performance differs greatly between runs and is very sensitive to the chosen initialization. The results also show that overly constraining the model by only allowing the model to only choose columns from the header and not from their mentions in questions (as in the pointer loss) can have negative impact on the model performance.

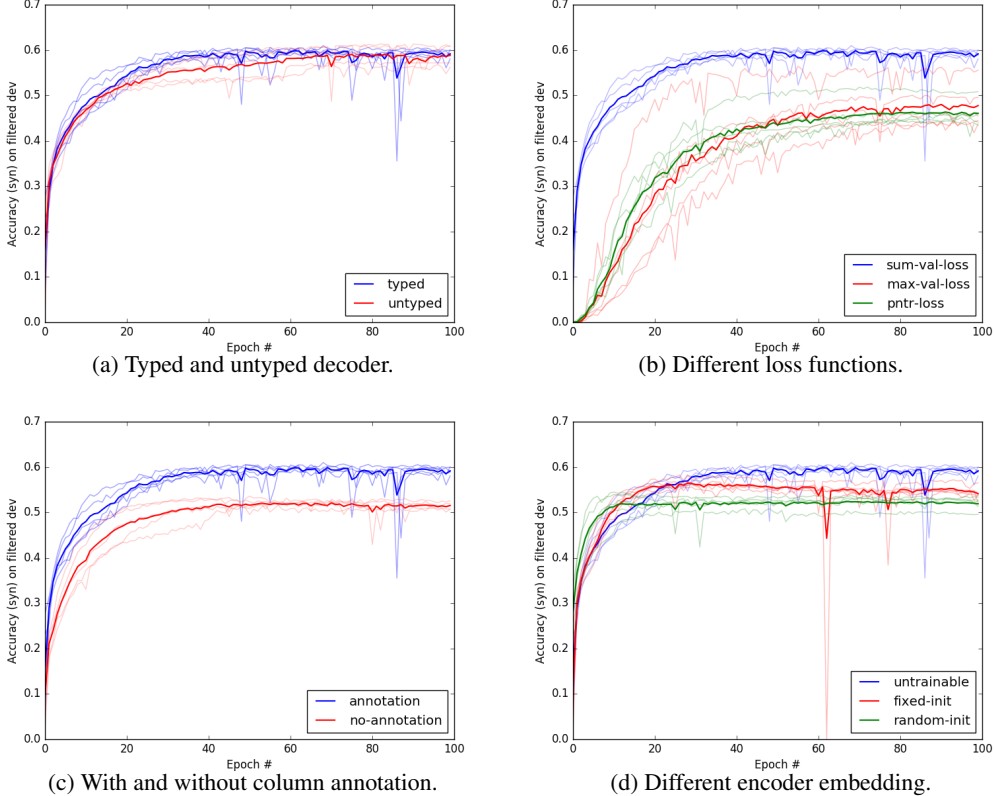

Figure 3: Ablation test results showing the syntax accuracy (on the filtered dev set) for each setting. For each setting, transparent lines show actual accuries for all 5 runs, and the none-transparent line highlights its average accuracy.

- *Column Annotation*: We study the effect of performing column annotation during preprocessing in Fig. 3c. We observe that the model accuracy drops by 7.5% if trained and tested on questions without column annotation. The result suggests that deterministically linking entities with their column can benefit the model and incorporating entity linking provides an important performance boost. On the other hand, the results indicate that typed decoding and the value-based loss function alone already reach ~52.5% accuracy on unannotated questions, beating the Seq2SQL baseline.

- *Embedding Method*: Finally, we study different input token embeddings in Fig. 3d: untrainable $n$-gram + GloVe embedding (untrainable in the plot), trainable embedding with $n$-gram + GloVe initialization (fixed-init) and trainable embedding with random initialization (random-init). Our results show that incorporating prior knowledge through untrainable embeddings can effectively prevent over-fitting.

## 4.4 ERROR ANALYSIS & LIMITATIONS

To better understand the source of erroneous results, we classify errors made by our model by the part of a query (aggregation function, select column, or predicates) that was incorrectly predicted. Among the 6,024 incorrectly predicted cases, 32.0% cases use a wrong aggregation function, 47.1% cases copied the wrong column name, and 51.1% cases contain mistakes in predicates (27.6% cases made multiple mistakes). Notably, most cases with wrong predicates are due to selecting a wrong column to compare to. Such cases are typically caused by the correct column name is not mentioned in the question (e.g., the questions contains 'best', but the respective column is called 'rank') or because multiple columns with similar names exist (e.g., 'team 1', 'team 2'). These errors suggest that the model lacks understanding of the knowledge presented in the table, and that embedding

the table content together with the question (Krishnamurthy et al., 2017; Yih et al., 2015) could potentially improve the model.

That our model does not support multiple pointer headers and no external vocabulary for decoding constant results in 13.1% wrong predictions (e.g., our model cannot generate 'intisar field' from 'field of intisar' in the question or generate 'score 4–4' from the question 'which team wins by 4–4?'), which suggests that extending the model with multiple constant pointers per slot or introducing an extra decoding layer for constant rewriting could potentially improve the model.

Finally, we do not directly train our model to learn syntactically different but semantically equivalent program. 62.2% among all wrong queries yield a run-time error or return `None` during execution. This suggests that training our model with an reinforcement loop to explicitly punish ill-formed queries and reward semantically equivalent ones (Zhong et al., 2017) could further improve results.

## 5 Related Work

**Semantic Parsing**   Nearest to our work, mapping natural language to logic forms has been extensively studied in natural language processing research (Zettlemoyer & Collins, 2012; Artzi & Zettlemoyer, 2011; Berant et al., 2013; Wang et al., 2015; Iyer et al., 2017; Iyyer et al., 2017). Dong & Lapata (2016); Alvarez-Melis & Jaakkola (2016); Krishnamurthy et al. (2017); Yin & Neubig (2017); Rabinovich et al. (2017) are closely related neural semantic parsers adopting tree-based decoding that also utilize grammar production rules as decoding constraints. However, our model foregoes the complexity of generating a full parse tree and never produces non-terminal nodes, and instead retains the simplicity of a sequence decoder. This makes it substantially easier to implement and train, as the sequence model requires no explicit controller for production rule selection. To our knowledge, our model is also the first to use target token type information to specialize the decoder to a mode in which it copies from a type-compatible, restricted set of input tokens.

**Pointer Networks**   Pointer and copy networks enhance RNNs with the ability to reuse input tokens, and they have been successfully used in interactive conversation (Gu et al., 2016), geometric problems (Vinyals et al., 2015) and program generation (Zhong et al., 2017). Our model differs from previous approaches in that we use types to explicitly restrict locations in the input to point to; furthermore, we developed a new training objective to handle pointer aliases.

**Program Induction / Synthesis**   Program induction (Reed & De Freitas, 2015; Neelakantan et al., 2016; Graves et al., 2014; Yin et al., 2015) aims to induce latent programs for question answering; on the other hand, program synthesis models (Zhong et al., 2017; Parisotto et al., 2016) aim to generate explicit programs and execute the program to obtain answer. Our model follows the line of neural program synthesis models and trains directly with question program pairs.

**Orthogonal Approaches**   Entity linking (Calixto et al., 2017; Yih et al., 2015; Krishnamurthy et al., 2017) is a technique used to link knowledge between the encoding sequence and knowledge base (e.g., table, document) in semantic parsing that is orthogonal to the neural encoder decoder model. This technique can potentially be used to address our limitation in our deterministic column annotation process. Besides, reinforcement learning (Zhong et al., 2017) allows the model to freely learn semantically equivalent solutions to user questions, and can be combined with our model to further improve its accuracy.

## 6 Conclusion

We presented a new sequence to sequence based neural architecture to translate natural language questions over tables into executable SQL queries. Our approach uses a simple type system to guide the decoder to either copy a token from the input using a pointer-based copying mechanism or generate a token from a finite vocabulary. We presented a sum-transfer value based loss function that transforms a distribution over pointer locations into a distribution over token values in the input to efficiently train the architecture. Our evaluation on the WikiSQL dataset showed that our model significantly outperforms the current state-of-the-art Seq2SQL model.

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

## A    APPENDIX

### A.1    EXAMPLES OF GENERATED QUERIES

We show a few examples that our model correctly and wrongly predicted collected from the evaluation result from the test dataset.

**Correct example (1)**

- Table: 1-1014206-2 [#, shipyard, laid down, launched, commissioned, fleet, status]
- Question: *"List the # for ships commissioned on september 30, 1967"*
- Solution: `Select # From` 1-1014206-2 `Where` commissioned = september 30, 1967

- Prediction: `Select` # `From` 1-1014206-2 `Where` commissioned = september 30, 1967

**Correct example (2)**

- Table: 2-17829169-2 [rank, gold, silver, bronze, total]
- Question: *"what is the most possible bronze medals when rank is more than 11 and there are fewer than 0 gold medals ?"*
- Solution: `Max` bronze `From` 2-17829169-2 `Where` rank > 11 `And` gold < 0
- Prediction: `Max` bronze `From` 2-17829169-2 `Where` rank > 11 `And` gold < 0

**Correct example (3)**

- Table: 2-18421908-1 [year, competition, venue, position, event]
- Question: *"what is the position for event discus in year 2013?"*
- Solution: `Select` position `From` 2-18421908-1 `Where` event = discus `And` year = 2013
- Prediction: `Select` position `From` 2-18421908-1 `Where` year = 2013 `And` event = discus

(Remarks: the colum name 'event' before 'event discus' is annotated to the question during column annotation since 'discus' uniquely appear in the event column.)

**Example with wrong target column**

- Table: 1-11206916-1 [team, stadium, capacity, highest, lowest, average]
- Question: *"what was the lowest highest attendance for the dumbarton team ?"*
- Solution: `Min` highest `From` 1-11206916-1 `Where` team = dumbarton
- Prediction: `Min` lowest `From` 1-11206916-1 `Where` team = dumbarton

(Remarks: The model fails recognize the target should be highest instead of 'lowest' that is used to specify the aggregation function `Min`.)

**Example with wrong aggregation function**

- Table: 2-18569335-2 [rank, heat, name, nationality, time]
- Question: *"how many heat did runners from nationality guinea-bissau run , with rank higher than 33 ?"*
- Solution: `Sum` heat `From` 2-18569335-2 `Where` nationality = guinea-bissau `And` rank < 33
- Prediction: `Count` heat `From` 2-18569335-2 `Where` nationality = guinea-bissau `And` rank > 33

(Remarks: The model fails to recognize that the 'heat' column stores aggregated value which should be count using `Sum`. Besides, the model does not recognize 'higher rank' indicates smaller numerical value.)

**Example with wrong predicate (1)**

- Table: 2-18661293-4 [rank, nation, gold, silver, bronze, total]
- Question: *"what 's the gold medal count for total nation with a bronze count more than 0 and a total less than 54 ?"*
- Solution: `Max` gold `From` 2-18661293-4 `Where` bronze > 0 `And` nation = total `And` total < 54
- Prediction: `Sum` gold `From` 2-18661293-4 `Where` bronze < 0 `And` total < 54

(Remarks: The model fails to learn the existence of the 'total' row in the table and miss the choice for nation.)

**Example with wrong predicate (2)**

- Table: 2-1223181-1 [year, entrant, chassis, engine, points]

- Question: *"what is the most points for a vehicle with a lola thl1, chassis later than year 1986?"*
- Solution: `Max` points `From` 2-1223181-1 `Where` chassis = lola thl1 `And` year > 1986
- Prediction: `Max` points `From` 2-1223181-1 `Where` year = lola thl1 `And` chassis = lola thl1 `And` year > 1986

(Remarks: The model fails by predicting a wrong column and constant combination.)

