# OpenReview forum: "Pointing Out SQL Queries From Text"
_ICLR.cc/2018/Conference — Reject_

### Official Review · AnonReviewer3 · 2017-11-26
**The contributions have already been done in the past**

**Rating:** 3
**Confidence:** 4

**Review:**

This paper proposes a model for solving the WikiSQL dataset that was released recently.

The main issues with the paper is that its contributions are not new.

* The first claimed contribution is to use typing at decoding time (they don't say why but this helps search and learning). Restricting the type of the decoded tokens based on the programming language has already been done by the Neural Symbolic Machines of Liang et al. 2017. Then Krishnamurthy et al. expanded that in EMNLP 2017 and used typing in a grammar at decoding time. I don't really see why the authors say their approach is simpler, it is only simpler because the sub-language of sql used in wikisql makes doing this in an encoder-decoder framework very simple, but in general sql is not regular. Of course even for CFG this is possible using post-fix notation or fixed-arity pre-fix notation of the language as has been done by Guu et al. 2017 for the SCONE dataset, and more recently for CNLVR by Goldman et al., 2017.

So at least 4 papers have done that in the last year on 4 different datasets, and it is now close to being common practice so I don't really see this as a contribution.

* The authors explain that they use a novel loss function that is better than an RL based function used by Zhong et al., 2017. If I understand correctly they did not implement Zhong et al. only compared to their numbers which is a problem because it is hard to judge the role of optimization in the results.

Moreover, it seems that the problem they are trying to address is standard - they would like to use cross-entropy loss when there are multiple tokens that could be gold. the standard solution to this is to just have uniform distribution over all gold tokens and minimize the cross-entropy between the predicted distribution and the gold distribution which is uniform over all tokens. The authors re-invent this and find it works better than randomly choosing a gold token or taking the max. But again, this is something that has been done already in the context of pointer networks and other work like See  et al. 2017 for summarization and Jia et al., 2016 for semantic parsing.

* As for the good results - the data is new, so it is probable that numbers are not very fine-tuned yet so it is hard to say what is important and what not for final performance. In general I tend to agree that using RL for this task is probably unnecessary when you have the full program as supervision.

---

> ### Author Response · Authors · 2018-01-04
> **Response**
>
> We thank the reviewer for the questions about the novelty of our contributions.
>
> Q. First claimed contribution of using typing at decoding time is not novel...previous works have used grammar (or CFG) at decoding time?
>
> We would like to emphasize that using a type system is quite different from using grammars. A grammar typically only describes the set of valid syntactic programs, whereas the type system can rule out certain classes of programs that are syntactically correct but violate certain type constraints. For example, consider the grammar for predicates of the form “columnName op constant”. Here, the grammar would allow program predicates such as Age < “USA”, whereas the type constraint can rule out that “<” operator can only be applied to integer values and the column’s type should also be integer. However, we agree that for this dataset the difference between type system and grammar is not that significant since the queries were generated using pre-defined templates.
>
> Q. Loss function is standard?
>
> Our value-based loss based on sum-transfer is not the same as using cross-entropy loss with uniform distribution over multiple gold tokens (pointers). Optimizing the loss function with uniform distribution over gold tokens (as done in Jia et al. 2016) would try to learn to predict similar probabilities over all the gold pointers, whereas instead our loss function first translates the pointers to their values and then sums up the values to learn a probability distribution over values as opposed to over indices.
>
> Please let us know if there are more clarifications that might be helpful to explain the differences with the previous works.

---

### Official Review · AnonReviewer1 · 2017-11-26
**I am not convinced that the contributions are significant**

**Rating:** 4
**Confidence:** 4

**Review:**

The paper claims to develop a novel method to map natural language queries to SQL. They claim to have the following contributions:

1. Using a grammar to guide decoding
2. Using a new loss function for pointer / copy mechanism. For each output token, they aggregate scores for all positions that the output token can be copied from.

I am confident that point 1 has been used in several previous works. Although point 2 seems novel, I am not convinced that it is significant enough for ICLR. I was also not sure why there is a need to copy items from the input question, since all SQL query nouns will be present in the SQL table in some form.  What will happen if we restrict the copy mechanism to only copy from SQL table.

The references need work. There are repeated entries for the same reference (one form arxiv and one from conference). Please cite the conference version if one is available, many arxiv references have conference versions.

Rebuttal Response: I am still not confident about the significance of contribution 1, so keeping the score the same.

---

> ### Author Response · Authors · 2018-01-04
> **Response**
>
> Thanks for the helpful comments and feedback.
>
> Q. Using a grammar to guide decoding is not novel?
>
> We would like to emphasize that using a type system is quite different from using grammars. A grammar describes the set of valid syntactic programs, whereas the type system can rule out certain classes of programs that are still syntactically correct. For example, consider the grammar for predicates of the form “columnName op constant”. Here, the grammar would allow program predicates such as Age < “USA”, whereas the type constraint can rule out that “<” operator can only be applied to integer values and the column’s type should also be integer. However, we agree that for this dataset the difference between type system and grammar is not that significant since the dataset contains only queries generated from a set of pre-defined templates.
>
> Q. Why copy from input question? Why not restrict copy mechanism to only SQL table?
>
> Since SQL tables can be quite large (and even for this dataset tables typically have tens of rows/columns), we only embed the column names of the tables and the input question for efficiency reasons. Besides, constants used in the query are commonly mentioned in questions raised by the user.

---

### Official Review · AnonReviewer2 · 2017-11-28

**Rating:** 7
**Confidence:** 4

**Review:**

This paper presents a neural architecture for converting natural language queries to SQL statements. The model utilizes a simple typed decoder that chooses to copy either from the question / table or generate a word from a predefined SQL vocabulary. The authors try different methods of aggregating attention for the decoder copy mechanism and find that summing token probabilities works significantly better than alternatives; this result could be useful beyond just Seq2SQL models (e.g., for summarization). Experiments on the WikiSQL dataset demonstrate state-of-the-art results, and detailed ablations measure the impact of each component of the model. Overall, even though the architecture is not very novel, the paper is well-written and the results are strong; as such, I'd recommend the paper for acceptance.

Some questions:
- How can the proposed approach scale to more complex queries (i.e., those not found in WikiSQL)? Could the output grammar be extended to support joins, for instance? As the grammar grows more complex, the typed decoder may start to lose its effectiveness. Some discussion of these issues would be helpful.
- How does the additional preprocessing done by the authors affect the performance of the original baseline system of Zhong et al.? In general, some discussion of the differences in preprocessing between this work and Zhong et al. would be good (do they also use column annotation)?

---

> ### Author Response · Authors · 2018-01-04
> **Response**
>
> We thank the reviewer for the helpful feedback and comments.
>
> Q. How can the proposed approach scale for more complex queries such as join?
>
> For more complex SQL structures (e.g., join, group by), we can not statically pre-determine the types of cells as there might be multiple number of non-determinisms in the query template. For example, consider a template for the join query such as  “select col from T (join T)* where (pred)*”, where we have two sets of non-determinisms -- one for variable number of Table names T and another one for variable number of predicates. However, once we have resolved the earlier non-determinism during the decoding process, we can still use the type system to guide decoding for the later choices. For example, once `where` is decoded in the above template,the tokens afterwards would be templated into `column op val` tuples and decoder types can still be applied.
>
> Q. How is the preprocessing done different from Zhong et al.?
>
> While we didn’t directly run Zhong et al. baseline on the preprocessed dataset, we compare the difference of with and without using preprocessing (column annotation) in our best model in Section 4.3 (Figure 3c) -- the results show that preprocessing does improve our model performance, but the ablation test demonstrates that the major improvement of the model comes from the improved loss function.

---

### Decision · Program_Chairs · 2018-01-29
**ICLR 2018 Conference Acceptance Decision**

**Decision:**

Reject

**Comment:**

The pros and cons of the paper can be summarized below:

Pro:
* The improvements afforded by the method are significant over baselines, although these baselines are very preliminary baselines on a new dataset.

Con
* There is already a significant amount of work in using grammars to guide semantic parsing or code generation, as rightfully noted by the authors, and thus the approach in the paper is not extremely novel.
* Because there is no empirical comparison with these methods, the relative utility of the proposed method is not clear.

As a result, I recommend that the paper not be accepted at this time.